# Evaluation on Seismic Performance of Beam-Column Joints of Fabricated Steel Structure with Replaceable Energy-Dissipating Elements

**Yuanqi Li and Binhui Huang \*** 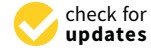

College of Civil Engineering, Tongji University, Shanghai 200092, China; liyq@tongji.edu.cn

\* Correspondence: keyhbh@163.com

**Abstract:** This research proposes a beam-column hinged joint with additional replaceable energy-dissipating elements, which is highly industrialized and fully fabricated. In this structure, steel beam and steel column are hinged with pins, at which corners a replaceable energy dissipation element is added. The energy dissipation element is rigidly connected to the steel column through a section of H-beam and high-strength bolts and is hinged to the steel beam using high-strength bolts. The main materials, such as energy dissipation elements, steel columns, and steel beams, are all steel with a design yield strength of 345 MPa. Under the condition that the vertical clear distance between the energy dissipation element and the steel beam is constant at 0.2 m, and the size and section of the beams and columns remain unchanged, six groups of different test samples are constructed by changing the thickness and the horizontal length of the energy-dissipating element. Through the experimental research and numerical simulation of 6 groups of specimens, the strength, stiffness, ductility, hysteresis curve, energy dissipation coefficient, equivalent viscous damping coefficient, and failure mechanism of the joints are obtained, and the horizontal section of the energy dissipation element is mainly analyzed. The effects of parameters such as the ratio of length to span and its ratio to the linear stiffness of steel beams on the seismic performance of the joints were compared with those of traditional welded steel frame beam-column joints. The research results show that the joints can be fully assembled, the energy-consuming components can be replaced, and the beam-column connection joints can be controlled in practical applications. The deviation between the experimental results and the numerical simulation results is less than 10%, which is in good agreement. The failure mode of the node conforms to the seismic performance concept of "energy-dissipating elements are destroyed first and easily replaced after earthquakes"; when the ratio of the horizontal length to the span of the energy-consuming components is 0.225, and the ratio to the linear stiffness of the steel beam is 0.7, the seismic performance is close to or superior to that of traditional welded steel frame beam-column joints, that is, equal to or better than traditional welded steel frame beam-column joints.

**Keywords:** replaceable energy-dissipating elements; fabricated steel structure; beam-column joints; seismic performance; linear stiffness

## 1. Introduction

The traditional steel frame is widely used in industrial and civil buildings due to its advantages of light weight, uniform material, high strength, good toughness, convenient production, and easy assembly, especially in earthquake-resistant fortified areas and high-rise buildings [1–3]. In this kind of structural system, beam-column joints often play an important role and are crucial to their ultimate bearing capacity and normal service capacity and even become an extremely important line of defense against structural collapse. In the traditional steel frame structure, some or all of the beam-column joints are manually welded on-site, and high-altitude operations occur from time to time. Although monitoring is required in terms of technology and management, it is difficult to control one by one. It

is inevitable and difficult to control the physical and psychological influence of operators, and the on-site environment is not transferred by human will, resulting in the insecurity of welding quality, and the final implementation effect of beam-column joints often deviates from the design. In the 1994 Northridge Earthquake in the United States and the 1995 Hyogoken–Nanbu Earthquake in Japan, through on-site inspection of buildings with steel frame structure system, the beam-column joints connected by welding showed various forms of damage or brittle fracture and some even collapsed due to their existence. It causes a lot of waste of resources and economic losses [4–6], and the restoration of buildings and urban functions are seriously affected. As a result, it has attracted the great attention of many scholars in the field of geotechnical engineering.

Based on the reduction or avoidance of on-site welding work, the use of fully assembled steel beam-column joints is a good choice, that is, through reasonable building structure design, the components are connected by high-strength bolts or rivets, etc. The strength, stiffness, and seismic performance meet the functional requirements of normal use, the quality of joints can be effectively controlled, and the on-site labor cost can be greatly reduced [7,8]. When the joint is damaged by a large earthquake, it is difficult to maintain and replace after the earthquake, and the building function is difficult to restore. Cheng-Chih Chen and other scholars proposed a fully assembled beam-column joint in which the components are basically connected by bolts, the welding between the components is completed in the factory, and the welding is basically eliminated on-site. Through experimental research and finite element numerical simulation, the results show that the possibility of brittle fracture can be effectively reduced, but the recoverable function after the earthquake is difficult to achieve [9–11]. Based on the recoverability after an earthquake, the joint researchers of the United States and Japan proposed a kind of "earthquake resilient structure", that is, a structure that can be restored to its use function after a small number of repairs or no services after an earthquake [12–14]. The earthquake-resilient structures are easy to construct and maintain and have high profitability in the whole life cycle. There are many ways to realize them, mainly including self-centering structures, rocking wall structures, and structures with replaceable elements [15–17], each of which has its own advantage. The self-centering or rocking wall structure releases the restraint between the superstructure and the foundation or the restraint between some components so that their mutual contact surfaces can bear the pressure but not the ability to be pulled. When subjected to earthquake action, the superstructure will sway and reset under the action of its own weight or prestress, reducing the demand for structural ductility due to earthquake action. The structure with replaceable components is to install energy-dissipating elements in relatively weak parts of the structure. When subjected to frequent earthquakes and fortification earthquakes, the energy dissipation elements work together with the main structure to contribute strength and stiffness to the structure; When subjected to a rare earthquake, the energy dissipating element yields first and dissipates energy. This process is similar to the action of a fuse to avoid damage or destruction to the main structure. After an earthquake, the damaged energy-consuming components can be easily and quickly removed and replaced. The implementation process has little impact on the normal use of the structure, thus realizing the recoverable function of the structure. Scholars such as Abazar [18,19] proposed a structure with replaceable components and conducted elastic and inelastic research and analysis on the collapse mechanism and ductility of the structure. The results show that the collapse mechanism and ductility of the structure are close to or better than traditional steel frames. However, the level of assembly and industrialization of the structure is relatively low. Scholars such as Hsu and Li [20–22] proposed a new structure with replaceable energy dissipation elements. Through experimental research and numerical simulation, the strength, yield mechanism, energy dissipation capacity, and hysteresis curve of the structure were analyzed. The results show that the failure mechanism of the structure is good and the seismic performance is excellent, but the connection between the steel beam and the steel column is rigid or semi-rigid with bolt connection, and the design still can be uncertain, which needs further in-depth research or enhanced.

At the same time, for a new type of structure with replaceable dissipative elements, when it is repaired after an earthquake, the new BMUA method proposed by Jice Zeng and Young Hoon Kim [23] can be used to update the structural stiffness by considering the coupled effect of mass and stiffness.

This paper presents a fully fabricated, beam-column hinged joint with replaceable dissipative elements. In the joint domain, the connection between the components is mainly as follows: the steel beam and the steel column are hinged with pins, and the energy dissipation element attached to the corner of the beam column is rigidly connected with the steel column by a section of H-beam and high-strength bolts and hinged with the steel beam with high-strength bolts. All components are made of Q345B [24], high-strength bolts are made of M12.9, and pins are made of 40Cr. There is no on-site welding in the node installation process, and the whole assembly is basically realized. The seismic energy dissipation mainly depends on the energy dissipation components, and it is easy to replace after the earthquake. The main research contents are as follows:

(1) Investigate the failure mechanism and the strength, stiffness, ductility, hysteresis curve, and energy dissipation capacity under external loads for the joints with replaceable energy-dissipating elements, fully fabricated, beam-column hinged.

(2) Investigate the stress distribution of the energy-dissipating elements for the joints with replaceable energy-dissipating elements, fully fabricated, beam-column hinged when part or all of the section of the energy dissipation element enters the yield state.

(3) Investigate the influence of the linear stiffness ratio of the energy dissipation element to the steel beam on the strength, stiffness, ductility, hysteresis curve, and energy dissipation capacity for the joints with replaceable energy-dissipating elements, fully fabricated, beam-column hinged when the position of the energy dissipation element is constant.

(4) Investigate the influence of the vertical position of the replaceable energy-dissipating element on the strength, stiffness, ductility, hysteresis curve, and energy dissipation capacity for the joints with replaceable energy-dissipating elements, fully fabricated, beam-column hinged when the vertical position of the replaceable energy dissipation element remains unchanged.

(5) Investigate the seismic performance differences between the joints with replaceable energy-dissipating elements, fully fabricated and traditional all-welded steel frame beam-column joints, and find out reasonable design principles or methods.

In the next investigation, when this new type of fabricated beam-column connection with replaceable energy dissipation elements is applied to actual projects, the PPSD method [25] will be used to evaluate the long-term behavior of the structure, using the TF method [26] to examine the effects of seismic or civil engineering equipment on the structure.

## 2. Experimental Program

### 2.1. Test Specimens

According to the main purpose of the experimental research, six groups of different steel frames were designed, numbered JD-1~JD-6. The common points of the six groups of steel frames are as follows: (1) The main dimensions of the steel frame are the same, including the height of 3.225 m, the span of 5 m, the net size of 0.200 m between the energy-dissipating element and the steel beam; (2) The main components of the steel frame are of the same type, 2 steel columns of H250 × 255 × 14 × 14, 1 steel beam of H250 × 125 × 6 × 9, 2 supports connected to the multifunctional structural test system, 2 L-shaped energy dissipation elements with a height of 0.150 m; (3) The main connection method of the steel frame is the same, in which the connection between the steel column and the bracket, the steel beam and the steel column is hinged. The energy dissipation element is rigidly connected to the steel column and hinged to the steel beam. The pins used in each group of steel frames are recycled, and the high-strength bolts are one-time use; (4) The main materials of the steel frame are the same, such as steel columns, steel

beams, energy dissipation elements, and connecting parts are Q345B, pins are 40Cr, and high-strength bolts are grade12.9. The differences between the 6 groups of steel frames are as follows: (1) The ratio of the horizontal length to the span of the energy dissipation element is different, such as 0.15 for JD-1, 0.2 for JD-2, 0.225 for JD-3~JD-5, and 0.25 for JD-6; (2) The thickness of the energy dissipation element is different, among which JD-1 is 0.014 m, JD-2 is 0.020 m, JD-3 is 16 mm, JD-4 is 0.022 m, JD-5 is 0.026 m, and JD-6 is 0.026 m. The main dimensions and connections of the 6 groups of steel frames are shown in Figure 1, and the physical parameters of the tested specimens are shown in Table 1.

**Table 1.** Average material physical parameters tested on specimens.

| Specimen | Length $H/L_b/L_{dg}$ (mm) | Specific Part | Yield Strength (MPa) | Young's Modulus E ($10^5$ MPa) | Linear Stiffness EI/L ($10^6$ N·m²/m) | Ratio of Linear Stiffness to Steel Beam |
|---|---|---|---|---|---|---|
| Steel Beam | 4530 | Flange | 397 | 2.04 | 1.59 | 1.00 |
| | | Web | 395 | | | |
| Steel column | 3225 | Flange | 396 | 2.05 | - | - |
| | | Web | 396 | | | |
| | 750 | JD-1 | 393 | 2.04 | 1.07 | 0.67 |
| | 1000 | JD-2 | 395 | 2.03 | 1.15 | 0.72 |
| Energy dissipating components | 1125 | JD-3 | 391 | 2.05 | 0.82 | 0.51 |
| | 1125 | JD-4 | 394 | 2.03 | 1.12 | 0.71 |
| | 1125 | JD-5 | 392 | 2.04 | 1.33 | 0.83 |
| | 1250 | JD-6 | 392 | 2.04 | 1.19 | 0.75 |

Remarks: H is the height of the steel frame, L is the span of the steel frame, $L_b$ is the length of the steel beam, and $L_{dg}$ is the horizontal length of the energy-dissipating element.

The steel beam of the steel frame is mainly determined by the vertical load and span, while the steel column is based on the seismic principle of "strong column and weak beam, strong shear and weak bending and strong joint" and the anti-seismic concept of "energy-consuming components are broken first, and can be replaced after the earthquake". The hinged connection between the steel column and the steel beam mainly needs to meet the following Equations (1) and (2), the connection between the energy dissipation element and the steel column should meet the following Equations (1)–(3) at the same time, and the bolts required should meet the bearing capacity requirements. The size of the energy-consuming components is analyzed and preliminarily determined according to the numerical simulation calculation. For the designed test samples, the energy dissipating elements are rigidly connected at one end and hinged at the other end. To avoid out-of-plane instability of energy dissipation elements, the slenderness ratio of nodes except JD-3 is less than 150, so the energy dissipation of plastic deformation mainly comes from in-plane. At the same time, to compare the seismic performance of the traditional all-welded steel frame, a group of steel frames is designed in which the steel beams and columns are the same as those in the test, and the beam-column joints are according to the equal-strength all-welding. In this paper, the steel frame will only be carried out through finite element numerical simulation analysis [27,28].

$$\Sigma W_{pc} f_{yc} \geq \eta \Sigma W_{pb} f_{yb} \tag{1}$$

$$M_\mu^j \geq \eta_j M_p \tag{2}$$

$$V_\mu^j \geq 1.2 \left( \sum M_p / l_n \right) \tag{3}$$

where $W_{pc}$ and $W_{pb}$ are the plastic section moduli of the columns and beams respectively; $f_{yc}$ and $f_{yb}$ are the steel yield strengths of the columns and beams respectively; $\eta$ is the strong column coefficient; $M_\mu^j$ is the ultimate flexural bearing capacity of the connection; $M_p$ is the plastic flexural bearing capacity of the beam; $\eta_j$ is the connection coefficient, which is 1.30 when the beam-column is welded, and 1.25 when the energy dissipation element

is connected with the short beam bolts; $V_\mu{}^j$ is the ultimate shear bearing capacity of the connection; $l_n$ is the net span of the beam [27].

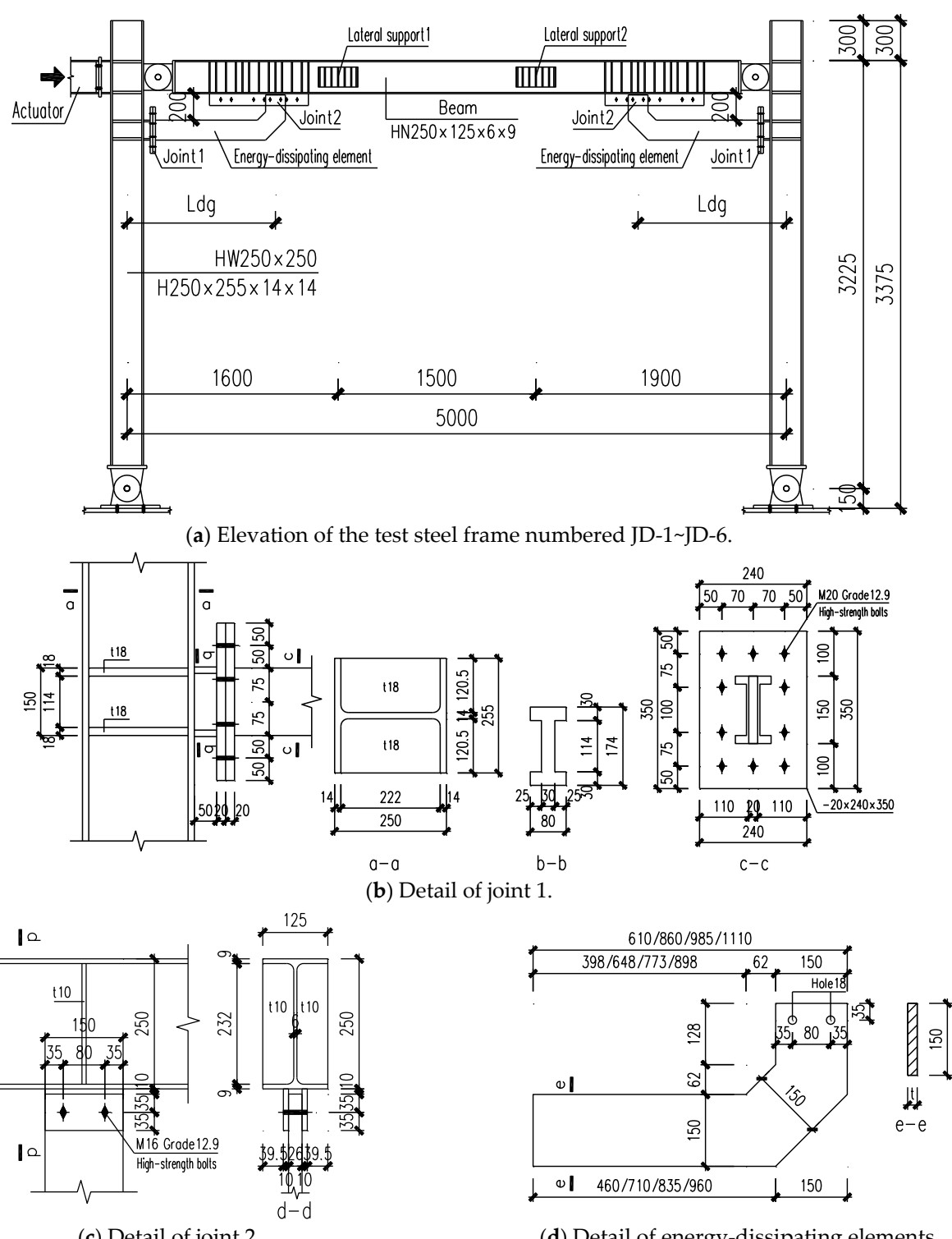

(**a**) Elevation of the test steel frame numbered JD-1~JD-6.

(**b**) Detail of joint 1.

(**c**) Detail of joint 2

(**d**) Detail of energy-dissipating elements.

**Figure 1.** Geometric dimensions and details of the test steel frame numbered JD-1~JD-6 (Unit: mm = $10^{-3}$ m).

### 2.2. Test Device and Loading System

The test adopts a quasi-static test device, which is mainly composed of a loading device, a support device, an observation device, and a safety device. The loading device is an electro-hydraulic servo system with a loading capacity of 300 kN and a range of 300 mm, which is not less than 1.5 times the calculated bearing capacity and ultimate deformation of the test frame, and is connected to the beam-column node on the left side of the steel frame; The test frame is firmly connected to the bottom of the test device through high-strength bolts, and the stiffness of the test support device is much greater than 10 times that of the bottom of it; The observation device is composed of strain gauges, strain gauges, computers and data acquisition systems installed on the steel frame to monitor the test status in real-time; The safety device is composed of a steel frame connected up and down with the multi-functional structural test system, and is connected with the test structure to prevent the lateral instability of the test frame. At the same time, for the steel frame to slide freely in the horizontal direction, sliding rollers are used to connect the steel frame at the joint, as shown in Figure 2. Considering that the replaceable energy dissipation element is connected to the steel column, which will have a certain impact on the calculated length of the steel column, the joint test is designed based on the full-scale model under the actual working state, and a steel frame is used to ensure that it is consistent with the actual design model.

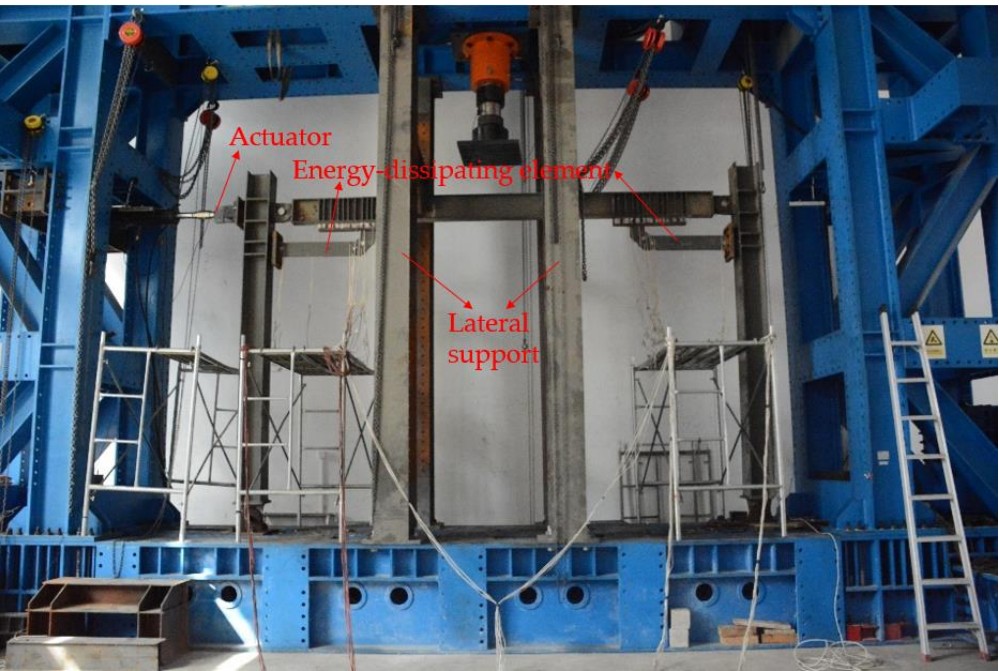

**Figure 2.** Test device.

The measurement of the test is determined according to the purpose of the test, and the main contents are as follows: the strain at the energy dissipation element and its connected components, the load-displacement curve of the steel frame, the load when the energy-dissipating element is damaged, etc. Strain measurement is an important content for analyzing structural failure mechanisms and stress distribution of main components. The strain gauge pasted on the test member is a foil resistance strain gauge with temperature self-compensation, which is BE120-5AA (11). The loading adopts the ZT-SAD200 actuator with a loading capacity of 200KN and a double spherical hinge at the end. The displacement is measured by the LVDT sensor, which is fixed on the tested steel frame by the bracket. In this test, four displacement gauges with a range of 0.300 m are installed, which are located on the outer side of the steel column (one on each side) and the beam-column joint (one on each side). The arrangement of strain, load, and displacement sensors is shown in Figure 3.

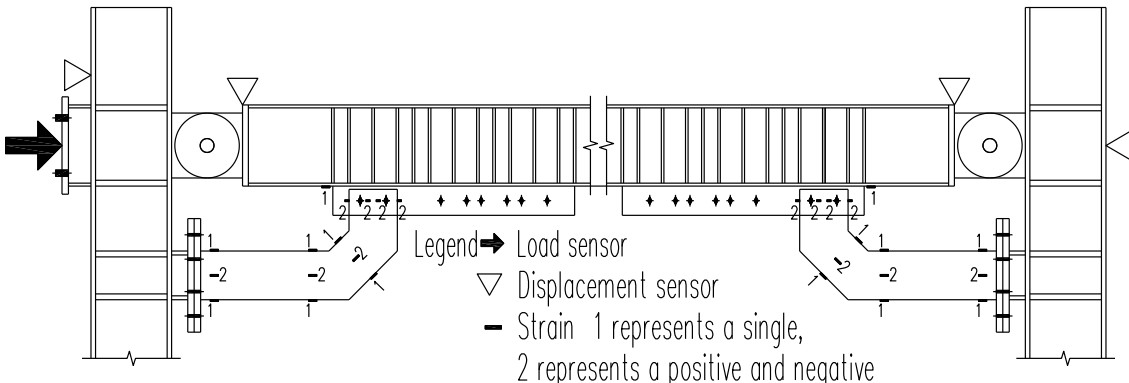

**Figure 3.** Sensors of strain, load, and displacement.

The test adopts the variable displacement loading method, as shown in Figure 4. Before the formal test, the loading value should not exceed 30% of the calculated value of the yield displacement load for preloading, and the test is repeated twice to check whether the test structure installation, loading equipment, measurement system, and test operators are ready. After the preloading is normal, the loading will be formally carried out according to the principle that the displacement loading of each stage is 5mm and the reciprocating cycle is 2 times before the structure yields; After the structure yields, it is loaded according to the principle of 10 mm displacement load per stage and two reciprocating cycles. When the loading value is close to the calculated value, and the stress on the upper edge or the middle of the energy dissipation element away from the column end is close to the yield strength, the test is terminated after 2 reciprocating cycles, and the test loading at this time is taken as the final level. The last stage loading displacement value of JD1 is 0.070 m, JD3 is 0.060 m, and the rest are 0.080 m.

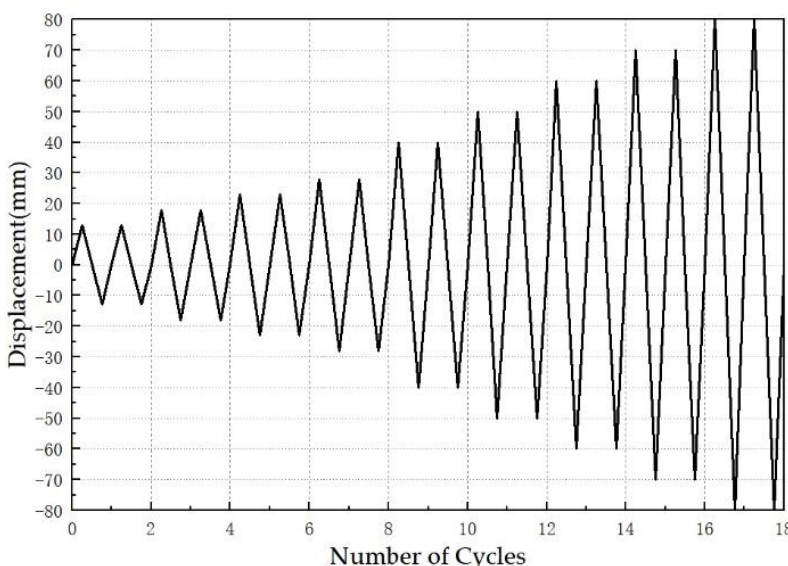

**Figure 4.** Loading method of variable displacement.

## 3. Failure Mode

Through reasonable design, the failure mode of traditional welded steel frame subjected to earthquake action is as follows: (1) When subjected to more than earthquake action, the steel frame is in an elastic state, bearing vertical and horizontal bearing capacity and deformation until reaching the elastic limit of the structure; (2) When the seismic action gradually increases, the steel beam flanges at the node domain begin to yield. For the traditional welded steel frame designed in this paper, this phenomenon occurs when the

horizontal displacement reaches about 1.2% of the storey height; (3) When the seismic action further increases or enters the rare earthquake action, the yield range of the steel beam section at the node domain expands until a plastic hinge is formed, and the energy dissipation behavior ends. For the traditional welded steel frame designed in this paper, this phenomenon occurs when the horizontal displacement reaches about 2% of the storey height. During the above process, the steel column is basically in an elastic state. Figure 5a shows the failure mode of the conventional steel frame structure.

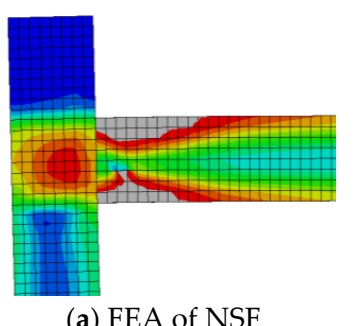

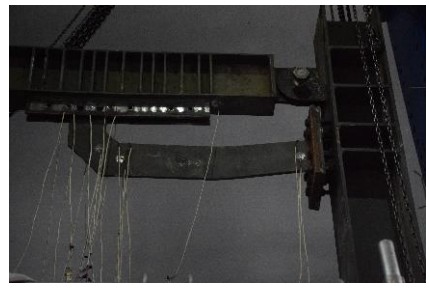

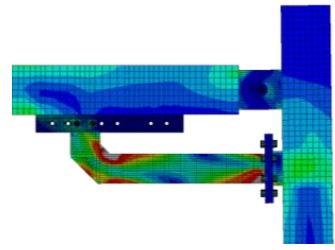

(**a**) FEA of NSF        (**b**) Experiment of JD-3        (**c**) FEA of JD-3

**Figure 5.** Experiment and FEA of NSF and JD-3.

The failure modes of the fully fabricated and beam-column hinged joints with replaceable energy dissipation elements under the action of earthquakes are significantly different, the energy dissipation capacity of which mainly comes from the energy dissipation elements. Its main working process is as follows: (1) under the action of frequent earthquakes, the energy dissipation element and the steel frame cooperate in providing elastic stiffness for the structure, bear vertical and horizontal loads and corresponding deformation; (2) with the gradual increase of the horizontal action, the deformation of the structure increases, but the deformation rate of the energy dissipation element is faster than that of other components, and the edge near one end of the steel column begins to yield and enters the energy dissipation stage. For the steel frame with replaceable dissipative elements designed in this paper, this phenomenon occurs when the horizontal displacement reaches about 0.4% of the storey height; (3) when the horizontal action continues to increase, the other side edge of the energy dissipation element in the horizontal direction yields successively, and also enters the plastic stage to dissipate energy; (4) when the horizontal action increases to a certain stage, the energy dissipation element yields at the section near one end of the steel column to form the first plastic hinge, and the behavior of energy dissipation is further obvious; (5) when the horizontal action is close to the limit load of the structure, or the earthquake is rare, the cross-section on the other side of the horizontal direction of the energy dissipation element continuously yields to form a second plastic hinge, and the behavior of energy dissipation ends. For the steel frame with replaceable dissipative elements designed in this paper, this phenomenon occurs when the horizontal displacement reaches about 2% of the storey height. After going through the above process under the horizontal action, except for the energy dissipation element, which has gone through the elastic-plastic stage, the rest of the components are always in the elastic state. Figure 5b,c shows the typical failure mode of JD-3 with a fabricated steel frame with replaceable dissipative elements, where gray indicates that the stress reaches a yield strength of 390 MPa.

## 4. Comparison of Results

### 4.1. Strength

Subjected to horizontal load, the traditional welded steel frame begins to be in an elastic working state until the stress of the steel beam flange in the node domain reaches the yield strength; as the horizontal load continues to increase, the yield range of the steel beam flange in the node domain continues to expand until the entire section yields and forms a

plastic hinge, and the steel frame enters the plastic limit state from the time. Subjected to the horizontal load, the assembled steel frame with replaceable energy dissipation element begins to be in an elastic state until the stress on the outer edge of the energy dissipation element near the steel column reaches the yield strength; as the horizontal load continues to increase, the steel frame is in the plastic limit state until a plastic hinge appears at both ends of the horizontal section of the energy dissipation element, and the rest of the components are always in an elastic state. They are shown in Table 2 with the elastic and plastic ultimate strengths of 6 groups of steel frames with replaceable energy dissipation elements and 1 group of traditionally welded steel frames. It can be seen from the table that the test results of the 6 groups of steel frames are very close to the results of the finite element numerical simulation. The elastic limit state is that the horizontal displacement is approximately 0.4% of the story height, and the plastic limit state is that the horizontal displacement is approximately 2% of the story height. The strength increases with the thickness and horizontal length of the energy-dissipating element. It is obvious that JD-1 and JD-3 are smaller than the other nodes. This situation is mainly caused by the fact that the neutral axis of the combined section formed by the energy dissipation element and the steel beam deviates to the side of the steel beam, and the yield range of the side is close to the steel column has a faster growth rate.

**Table 2.** Strength of the specimens.

| Joint Type | Elastic Ultimate Strength | | | | %EUS | Plastic Ultimate Strength | | | | %PUS |
|---|---|---|---|---|---|---|---|---|---|---|
| | Experiment | | FEA | | | Experiment | | FEA | | |
| | Dis (mm) | F (KN) | Dis (mm) | F (KN) | | Dis (mm) | F (KN) | Dis (mm) | F (KN) | |
| JD-1 | 14.50 | 28.84 | 14.22 | 29.46 | 2.15 | 68.84 | 92.40 | 69.68 | 96.01 | 3.91 |
| JD-2 | 14.50 | 32.52 | 14.50 | 33.22 | 2.15 | 79.00 | 112.06 | 79.83 | 121.12 | 8.08 |
| JD-3 | 14.50 | 30.45 | 14.50 | 31.57 | 3.68 | 59.05 | 84.47 | 59.04 | 88.12 | 4.32 |
| JD-4 | 14.50 | 33.37 | 14.50 | 34.64 | 3.81 | 79.00 | 121.79 | 79.08 | 130.54 | 7.18 |
| JD-5 | 14.50 | 34.73 | 14.50 | 37.31 | 7.43 | 79.70 | 133.14 | 79.79 | 139.01 | 4.41 |
| JD-6 | 14.50 | 34.42 | 14.50 | 34.95 | 1.54 | 79.00 | 129.21 | 79.08 | 135.35 | 4.75 |
| NSF | - | - | 39.29 | 53.39 | - | - | - | 98.22 | 108.08 | - |

Remarks: Dis is the horizontal displacement; F is the horizontal force; Experiment is the test result; FEA is the numerical analysis result; NSF is the traditional welded joint of the steel frame.

### 4.2. Stiffness

When the traditional welded steel frame is subjected to horizontal load, the stiffness of the steel beam basically maintains a consistent elasticity before it enters the yield stage; When the steel beam enters the elastic-plastic stage, the stiffness gradually degenerates. When the assembled steel frame with replaceable energy dissipation element is subjected to horizontal load, the elastic behavior of the stiffness of the energy dissipation element is basically stable before the energy dissipation element enters the yield stage. The degradation is more pronounced as both sides of its horizontal segment enter the yield phase. They are shown in Table 3 with the elastic stiffnesses of 6 groups of steel frames with replaceable energy dissipation elements and 1 group of traditionally welded steel frames. It can be seen from the table that the test results of the 6 groups of steel frames are very close to the results of the finite element numerical simulation. The elastic stiffness increases with the thickness and horizontal length of the dissipative element and outperforms traditionally welded steel frames.

**Table 3.** The stiffness of the specimens.

| Joint Type | Elastic Stiffness (kN/m) | | |
|---|---|---|---|
| | Experiment | FEA | %ES |
| JD-1 | 2.00 | 2.04 | 2.00 |
| JD-2 | 2.21 | 2.26 | 2.26 |
| JD-3 | 2.08 | 2.15 | 3.37 |
| JD-4 | 2.27 | 2.31 | 1.76 |
| JD-5 | 2.36 | 2.43 | 2.97 |
| JD-6 | 2.35 | 2.44 | 3.83 |
| NSF | - | 1.36 | - |

*4.3. Ductility*

The ductility coefficient of a traditional welded steel frame is the ratio of the maximum deformation $\mu_m$ in the elastic-plastic phase of the structure to the maximum deformation $\mu_y$ in the elastic phase [29]. Based on the seismic design concept of "energy dissipation elements are destroyed first and can be replaced after earthquakes", when both ends of the horizontal section of the energy dissipation elements yield and the remaining components and their connectors are in an elastic state, the maximum deformation of assembled steel frame with replaceable energy dissipation elements is $\mu_m$, and the ductility coefficient is the ratio of $\mu_m$ to the maximum deformation $\mu_y$ of all components and their connectors in an elastic state. They are shown in Table 4 with the ductility of 6 sets of steel frames with replaceable energy dissipation elements and 1 set of conventional welded steel frames. It can be seen from the table that the test results of the 6 groups of steel frames are very close to the results of the finite element numerical simulation. When the ductility increases with the horizontal length and thickness of the energy dissipating element, it first increases and then decreases, Among which, the ductility of JD-1 and JD-3 are smaller, and the rest are basically close, and their performance is better than that of the traditional welded steel frame.

**Table 4.** Ductility of the specimen.

| Joint Type | Ductility | | |
|---|---|---|---|
| | Experiment | FEA | % Ductility |
| JD-1 | 4.84 | 4.90 | 1.24 |
| JD-2 | 5.45 | 5.51 | 1.10 |
| JD-3 | 4.07 | 4.07 | 0 |
| JD-4 | 5.45 | 5.45 | 0 |
| JD-5 | 5.50 | 5.50 | 0 |
| JD-6 | 5.45 | 5.45 | 0 |
| NSF | - | 2.50 | - |

*4.4. Hysteresis Curve*

The hysteresis curve of traditional welded steel frame joints has the characteristics of obvious shuttle shape, high plumpness, good plastic deformation ability, and strong energy dissipation ability [6,30,31]. Analyzing from 6 groups of assembled steel frame nodal tests with replaceable energy dissipation elements, the hysteresis curves are basically shuttle-shaped and relatively full, with good stability performance, which is highly similar to the results of finite element numerical simulation. When the energy dissipation element is in the elastic stage, the stiffness enhancement or degradation of the structure basically does not occur under the same level of displacement cyclic loading; When the energy dissipating element enters the plastic stage until the sections on both sides of the horizontal section yield, the slope of the loading and unloading curves at the same level of displacement decreases with the increase of the number of cycles, and the energy dissipation behavior shows a better level, among which JD-1 The displacement plastic limit load of JD-3 and

JD-3 is smaller, the energy dissipation behavior of JD-5 and JD-6 is not sufficient, and the energy dissipation behavior of JD-2 and JD-4 is relatively good, but the hysteresis curve of JD-4 is much closer to the traditional welded steel frame. They are shown in Figure 6 with the hysteresis curves of 6 groups of steel frames with replaceable energy dissipation elements and 1 group of traditionally welded steel frames.

### 4.5. Energy Dissipation

The dissipative capacity of the beam-column joint of the steel frame is calculated using the area enclosed in the load-displacement hysteresis curve, as shown in Figure 7. It was measured by the energy dissipation coefficient E and the equivalent viscous damping coefficient $\zeta_{eq}$ [29] for the evaluation of 6 groups of steel frames with replaceable energy dissipating elements and 1 group of traditionally welded steel frames, according to Equations (4) and (5). The results are shown in Table 5. It can be seen from the table that the results of the test are highly similar to the finite element numerical simulation, in which JD-2 is larger, JD-5 and JD-6 are smaller, and the rest are moderate, but JD-4 is closer to the traditional welded steel frame.

$$E = \frac{S_{(ABC+CDA)}}{S_{(OBE+ODF)}} \tag{4}$$

$$\zeta_{eq} = \frac{1}{2\pi} \bullet \frac{S_{(ABC+CDA)}}{S_{(OBE+ODF)}} \tag{5}$$

where $S_{(ABC+CDA)}$—the area enclosed in the figure of the hysteresis curve; $S_{(OBE+ODF)}$—the sum of the areas of the triangles OBE and ODF in the figure.

**Table 5.** Energy dissipation coefficient and equivalent viscous damping coefficient of the specimen.

| Joint Type | Energy Dissipation Coefficient E | | | Equivalent Viscous Damping Coefficient $\zeta_{eq}$ | | |
|---|---|---|---|---|---|---|
| | Experiment | FEA | %E | Experiment | FEA | %$\zeta_{eq}$ |
| JD-1 | 0.86 | 0.87 | 1.16 | 0.14 | 0.14 | 0 |
| JD-2 | 0.95 | 0.97 | 2.11 | 0.15 | 0.15 | 0 |
| JD-3 | 0.87 | 0.87 | 0 | 0.14 | 0.14 | 0 |
| JD-4 | 0.87 | 0.89 | 2.30 | 0.14 | 0.14 | 0 |
| JD-5 | 0.64 | 0.64 | 0 | 0.10 | 0.10 | 0 |
| JD-6 | 0.67 | 0.67 | 0 | 0.11 | 0.11 | 0 |
| NSF | - | 0.744 | - | - | 0.12 | - |

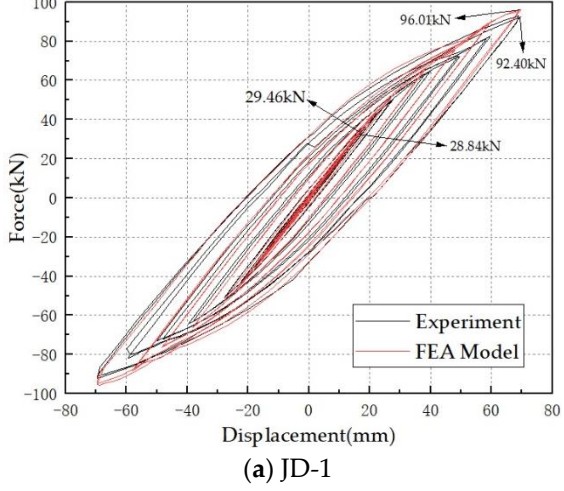

**(a)** JD-1

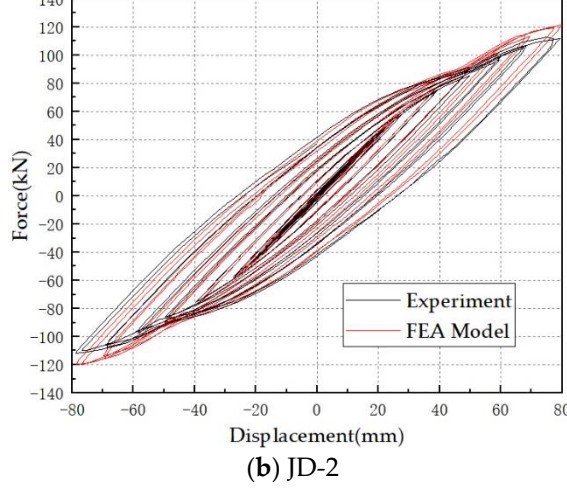

**(b)** JD-2

**Figure 6.** *Cont.*

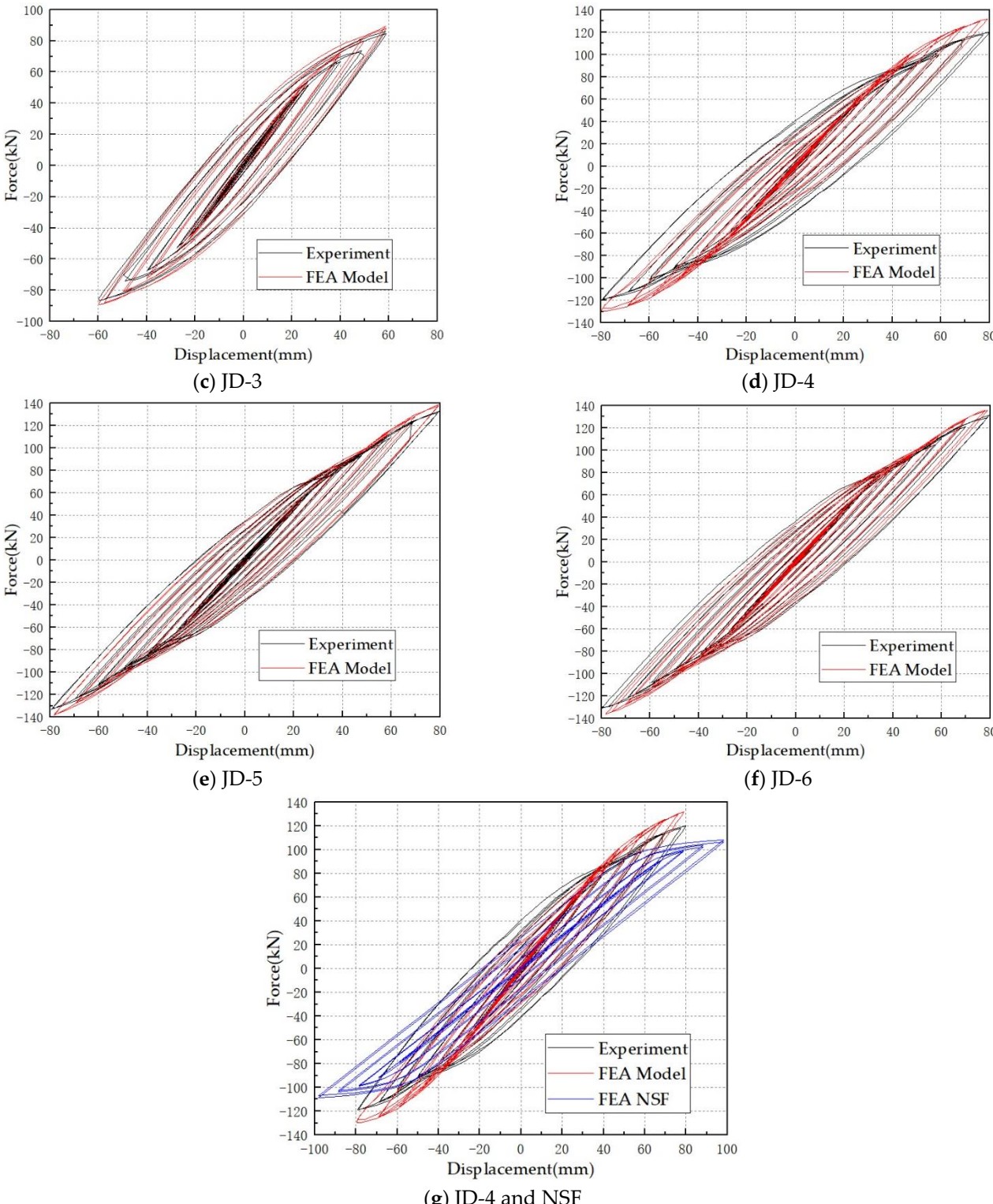

**Figure 6.** Hysteresis curve of beam-column joint test and FEA.

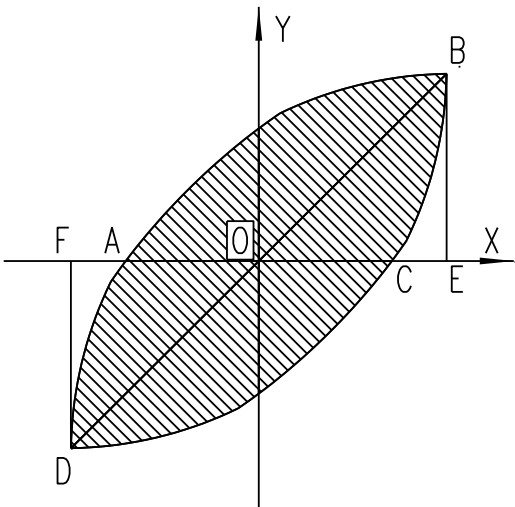

**Figure 7.** Area diagram of the load-displacement hysteresis curve.

## 5. Numerical Verification

### 5.1. FEA Model

For 6 groups of steel frames with replaceable energy dissipation elements and 1 group of traditionally welded steel frames, the finite element numerical simulation adopts the elastic-plastic analysis method in the ABAQUS, in which geometric nonlinearity is also considered. The model is shown in Figure 8. According to the physical properties and geometric dimensions of the material, three sets of test specimens were taken for each test. The material constitutive relation adopts the tri-polyline model, the main parameters of which are obtained by the statistics of the tensile test. The data are mainly as follows: The yield strength of high-strength bolts is $f_y$ = 1080 MPa, the ultimate strength $f_u$ = 1200 MPa, the ultimate strain $\varepsilon_u$ = 0.2, the failure strength fb = 860 MPa, and the failure strain $\varepsilon_b$ = 0.5; The yield strength of the 40Cr pin is $f_y$ = 785 MPa, the ultimate strength $f_u$ = 980 MPa, the ultimate strain $\varepsilon_u$ = 0.2, the failure strength fb = 560 MPa, and the failure strain $\varepsilon_b$ = 0.5; The yield strength of energy dissipation element, steel column, steel beam and other connectors is $f_y$ = 390 MPa, ultimate strength $f_u$ = 540 MPa, ultimate strain $\varepsilon_u$ = 0.2, failure strength $f_b$ = 270 MPa, failure strain $\varepsilon_b$ = 0.5. The Young's modulus of each material was taken as 2.05e5 MPa, and the Poisson's ratio was taken as 0.3. The S4R element is applied to steel beams and columns, and the C3D8R element is used for the remaining components, with a total number of about 304,000 elements. The contact relationship between the assembly components and accessories is mainly as follows: Tie contact is used between energy dissipation elements, steel columns, steel beams, and their connecting plates, in which the master and slave surfaces are determined in the order of large contact area, high stiffness, and sparse mesh [32]; the friction and hard contact are adopted between the connecting plate and the high-strength bolts according to the general contact method, in which the friction coefficient is 0.3, the pre-tension force of M20 high-strength bolts is 155 kN, and the pre-tension force of M16 high-strength bolts is 100 kN; the hard contact is applied between the connector, and the 40 Cr pin shaft adopts, in which the friction coefficient is 0, and the bolt pretension is 0 kN. The loading method is basically the same as the test. The boundary conditions of the base plate of the column base are carried out according to the rigid connection.

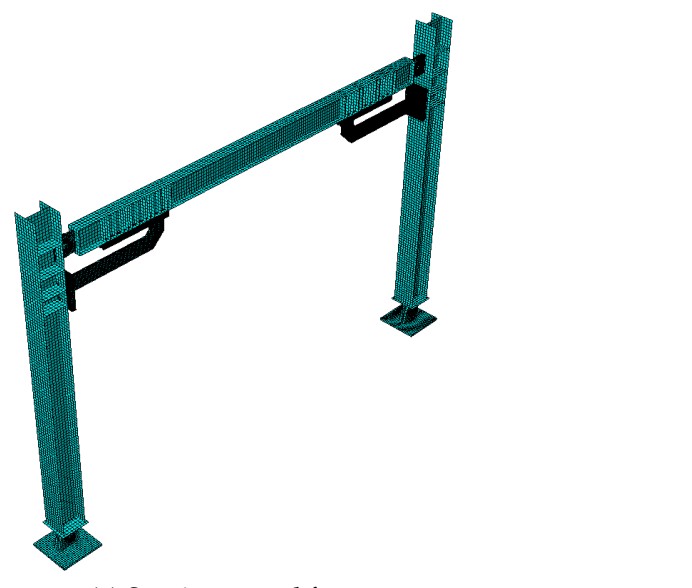

(**a**) Specimen steel frame          (**b**) Traditional steel frame

**Figure 8.** FEA model.

*5.2. Stress*

Based on the seismic design concept of "energy dissipating elements are destroyed first and can be replaced after earthquakes", when the six groups of test steel frames reach the plastic limit, the stress distribution of the energy dissipating elements is shown in Figure 9, in which The gray indicates that the stress reaches the yield strength of 390 MPa, and the other colors indicates unyielding. The results show that: (1) when the ratio of the linear stiffness of the energy dissipation element to the steel beam is less than 0.7, the energy dissipation behavior is more sufficient, but the contribution to the strength and stiffness of the steel frame is small; (2) when the ratio of their linear stiffness is greater than 0.7, the energy dissipation behavior is insufficient, but the contribution to the strength and stiffness of the steel frame is too large; (3) when the ratio of their linear stiffness is about 0.7, their energy dissipation behavior is sufficient, and their contribution to the strength and stiffness of the steel frame is appropriate.

*5.3. Hysteretic Behavior*

The comparison of load-displacement curves calculated from experiments and numerical simulations are shown in Figure 6. The results show that the curves in the two cases are basically close, which verifies the adaptability of the results, and the error is within 10%, which is caused by the accumulation of errors in production and experimental operation.

*5.4. Failure Mode*

The comparison mode of nodal failure calculated by experiment and numerical simulation is shown in Figure 5. The results show that the trend of the destruction mode is basically consistent, which verifies the correctness of the node destruction mode. The damage of 6 groups of steel frames with replaceable energy-dissipating elements is mainly concentrated on the energy-dissipating elements, and the remaining components and accessories basically maintain elastic behavior during the whole process, which basically meets the objectives of the experimental research, while the damage of traditionally welded steel frames is mainly concentrated on the plastic hinge formation process at the end of the steel beam.

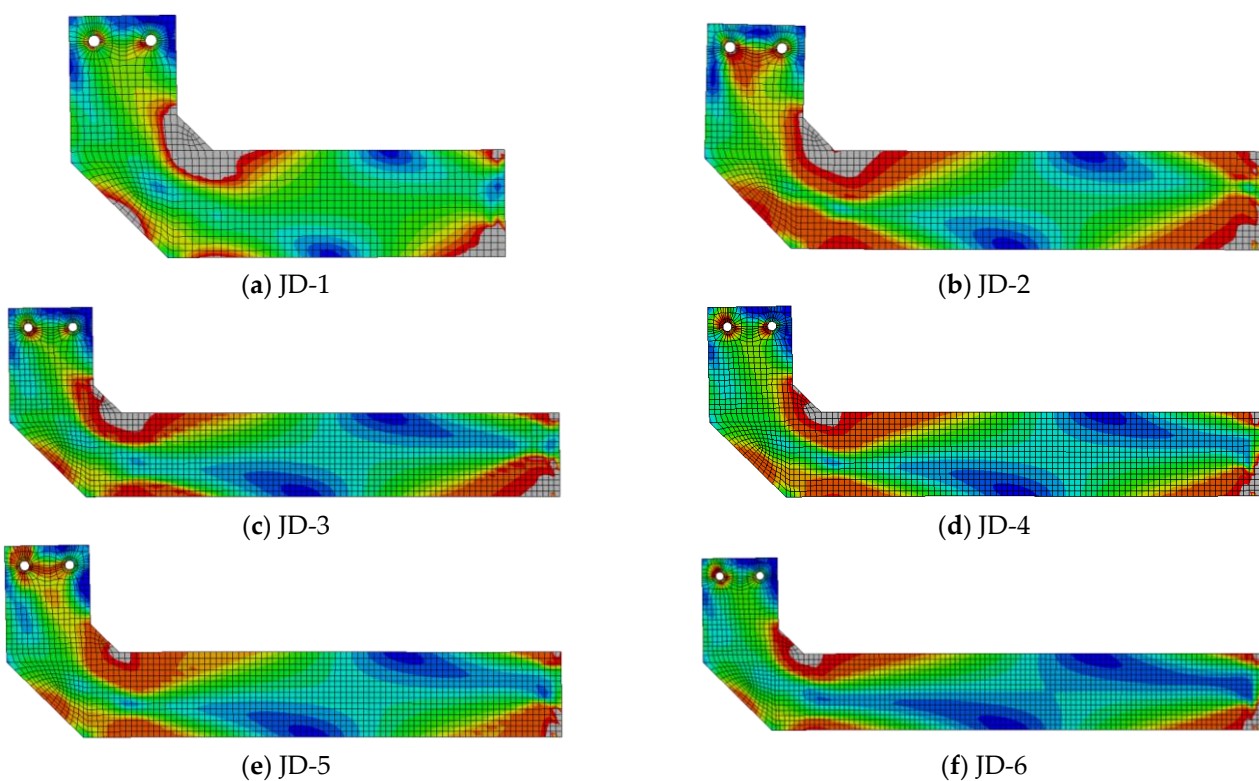

**Figure 9.** Plastic damage stress distribution diagram of energy dissipation element.

### 5.5. Energy Consumption Capacity

Table 6 shows the equivalent damping coefficients of the displacement ratios between 6 groups of steel frames with replaceable energy dissipation elements and 1 group of traditionally welded steel frames under various interlayer displacement ratios. The results show that the maximum equivalent damping coefficient of the steel frame with replaceable energy-dissipating element is basically greater than or close to 0.1, in which JD-1, JD-2, and JD-4 are close to the traditional welded steel frame, and the rest are small. JD-1 and JD-3 are larger when the interlayer displacement is relatively small, reflecting that their sufficient energy dissipation is concentrated in the early elastic-plastic stage; JD-2 and JD-4 are larger when the interlayer displacement is larger, which reflects that their sufficient energy dissipation is concentrated in the entire elastic-plastic process; JD-5 and JD-6 are larger in the case of the maximum interlayer displacement, reflecting that their energy dissipation capacity is not sufficient in the elastic-plastic process; Although the failure of 6 groups of steel frames with replaceable energy dissipation elements occurs at a level that the interlayer displacement is smaller than that of traditionally welded steel frames, it is basically larger than 2% of the rare earthquake range. The seismic energy-dissipating capacity of JD-4 is closer to the traditional welded steel frame.

**Table 6.** Equivalent viscous damping coefficient of the specimens.

| Joint Type | Equivalent Damping Coefficient under Various Interlayer Displacement Ratios | | | | | | |
|---|---|---|---|---|---|---|---|
| | 1.24% | 1.55% | 1.86% | 2.17% | 2.48% | 2.79% | 3.10% |
| JD-1 | 0.063 | 0.093 | 0.103 | 0.138 | - | - | - |
| JD-2 | 0.055 | 0.085 | 0.096 | 0.117 | 0.154 | - | - |
| JD-3 | 0.078 | 0.106 | 0.138 | - | - | - | - |
| JD-4 | 0.020 | 0.053 | 0.071 | 0.089 | 0.141 | - | - |
| JD-5 | 0.044 | 0.0625 | 0.076 | 0.080 | 0.102 | - | - |
| JD-6 | 0.040 | 0.062 | 0.078 | 0.087 | 0.107 | - | - |
| NSF | - | - | - | 0.01 | 0.048 | 0.085 | 0.118 |

When 6 groups of steel frames with replaceable energy dissipation elements and 1 group of traditionally welded steel frames are in the elastic-plastic stage, the accumulated plastic energy consumption is shown in Figure 10 under different horizontal displacement loads. The results show that JD-1 and JD-3 are smaller, and the rest are larger. When the horizontal length of the energy-dissipating element remains the same, the energy-dissipating capacity increases with its size, reflecting that the energy-dissipating capacity is affected by the factors. In 6 sets of steel frames with replaceable dissipative elements, the JD-4 performs well and is closer to the conventional welded steel frame.

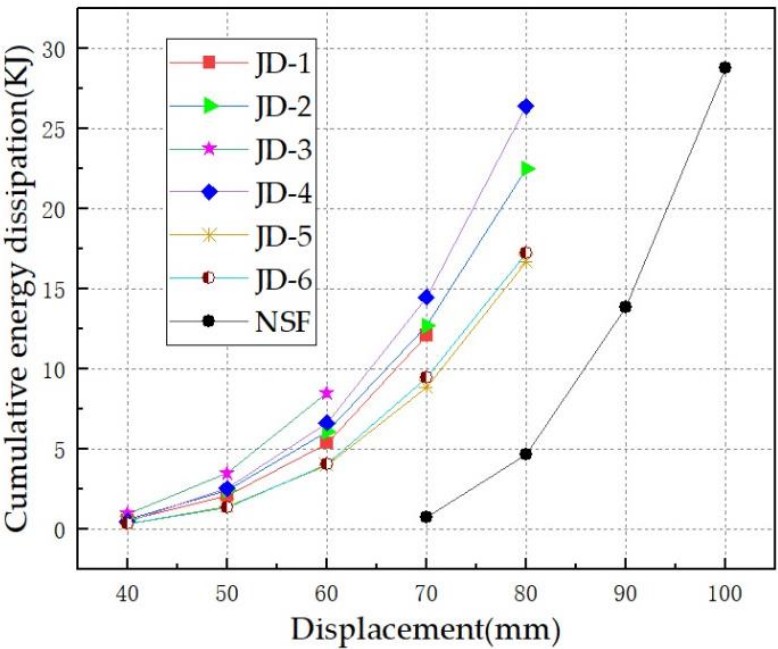

**Figure 10.** Cumulative energy dissipation of JD1~5 and NSF.

## 6. Conclusions

The research was focused on the main seismic performance evaluation factors such as failure mode, strength, stiffness, ductility, hysteresis curves, and energy dissipation capacity through the experimental study and finite element numerical simulation of 6 groups of steel frames with replaceable energy dissipation elements, and compared with the finite element numerical simulation of the traditional welded steel frame. The conclusions are as follows:

(1)  For the assembled and beam-column hinged joints with replaceable energy-dissipating elements, the seismic performance evaluation factors of the test and finite element numerical simulation such as the strength, stiffness, ductility, hysteresis curve energy dissipation coefficient, and equivalent damping coefficient are approximately consistent, which verifies that the test results are basically reliable.

(2)  For the assembled and beam-column hinged joints with replaceable energy-dissipating elements, as the failure process obtained by the test is consistent with the finite element numerical simulation in general, their destruction mode can realize the new seismic design concept of "energy consumption destruction first, and replacement after earthquake". Subjected to the action of frequent earthquakes, the energy dissipating elements cooperate with the rest of the structure in elastic behavior and provide sufficient strength and stiffness; Subjected to the action of moderate earthquakes and rare earthquakes, the energy-dissipating elements enter the plastic stage, providing sufficient energy-dissipating capacity, while the remaining components and accessories still maintain elastic behavior.

(3) For the assembled and beam-column hinged joints with replaceable energy-dissipating elements, when the horizontal length is constant, and the ratio of the linear stiffness of the energy dissipation element to the steel beam is approximately 0.7, their seismic performance is in a relatively good state. From the analysis of the 6 groups of tests, it can be concluded that the seismic performance of JD-2 and JD-4 is better than the other groups.

(4) For the assembled and beam-column hinged joints with replaceable energy-dissipating elements, under the condition of constant vertical position and the material properties of energy dissipation elements and steel beams, when the ratio of horizontal length to span is 0.225, their seismic performances are superior. From the analysis of 6 groups of tests, it can be concluded that the seismic performance of JD-4 is better than that of JD-2.

(5) For the assembled and beam-column hinged joints with replaceable energy-dissipating elements, compared with the traditional welded beam-column joints in the steel frame, when the ratio of linear stiffness of energy dissipation element to steel beam is about 0.7, and the ratio of horizontal length to span is about 0.225, they are considered to be a good alternative of it, because the seismic performance is basically close to or beyond that of traditional welded steel frame beam-column joints. From the 6 groups of test analysis, it can be concluded that the seismic performance of JD-4 is better than any other group, and it is closer to the beam-column joint of the traditional welded steel frame.

(6) For the assembled and beam-column hinged joints with replaceable energy-dissipating elements, their welding works can be completed in the factory, and there is basically no welding on site, which is conducive to environmental protection; Their beam-column connection nodes are hinged, which is completely controllable in practical applications compared to rigid or semi-rigid connection nodes connected by bolts; after the earthquake damage, the damaged energy-consuming components can be removed to restore normal use, which is convenient and fast and is conducive to improving the resilience of the city.

**Author Contributions:** Conceptualization, Y.L. and B.H.; methodology, Y.L. and B.H.; software, B.H.; validation, Y.L. and B.H.; formal analysis, B.H.; investigation, B.H.; resources, Y.L.; data curation, B.H.; writing—original draft preparation, B.H.; writing—review and editing, Y.L.; visualization, B.H.; supervision, Y.L.; project administration, Y.L.; funding acquisition, Y.L. All authors have read and agreed to the published version of the manuscript.

**Funding:** This research was funded by the National Natural Science Foundation of China grant number 51978500 and the APC was funded by the Author named Binhui Huang.

**Institutional Review Board Statement:** Not applicable.

**Informed Consent Statement:** Not applicable.

**Data Availability Statement:** Data is contained within the article.

**Acknowledgments:** The writers gratefully acknowledge the support for this work, which was funded by the National Natural Science Foundation of China (Grant Nos. 51978500).

**Conflicts of Interest:** The authors declare no conflict of interest.

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
