# Peer review of "Evaluation on Seismic Performance of Beam-Column Joints of Fabricated Steel Structure with Replaceable Energy-Dissipating Elements"

_sustainability, doi:10.3390/su14063350_

Round 1
Reviewer 1 Report
The presented work is an important topic for structural engineers to understand the performance of the replaceable energy-dissipating elements for structural safety and reliability. The paper presents the experimental and analytical results of the systems with varied joint designs. The data can be important for future research and the direction of the refined design. However, some clarification is needed in the result section.
Lines 88-98: Need to add references next to each author. Remove "Asghari et al. or two authors' last names" Please check the journal's style.
Line 105: What is Q345B? Isn't it Chinese Standard? Need to add a reference or standard specification. For example, the ASTM standard is used for materials fabricated or used in the U.S.
Line 99: Need to recognize some work relevant to recent research for the effort in simulation. There are some efforts to update the model's parameter in in-situ conditions (fields), where repair and replacement are needed.
Jice Zeng and Y.H. Kim, "Stiffness Modification-Based Bayesian Finite Element Model Updating to Solve Coupling Effect of Structural Parameters: Formulations" https://doi.org/10.3390/app112210615
Line 323: Currently, there is no experimental data regarding joint Type NSF. Is there any estimation from the design equation for the control system's performance?
Figure 6: Caption in Figure 6 shows Bs instead of experimental and analytical results. Need to update.
The authors need to add additional figures to show the elastic yielding point and ultimate strength point with one of the hysteresis curves.
Author Response
Response to Reviewer 1 Comments
Point 1: The presented work is an important topic for structural engineers to understand the performance of the replaceable energy-dissipating elements for structural safety and reliability. The paper presents the experimental and analytical results of the systems with varied joint designs. The data can be important for future research and the direction of the refined design. However, some clarification is needed in the result section.
Response 1: Make the following additions in the conclusion(6 conclusion). It has been modified as follows:
(1) For the assembled and beam-column hinged joints with replaceable energy-dissipating elements, the seismic performance evaluation factors of the test and finite element numerical simulation such as the strength, stiffness, ductility, hysteresis curve energy dissipation coefficient and equivalent damping coefficient are approximately consistent, which verifies that the test results are basically reliable.
(2) For the assembled and beam-column hinged joints with replaceable energy-dissipating elements, as the failure process obtained by the test is consistent with the finite element numerical simulation in general, their destruction mode can realize the new seismic design concept of "energy consumption destruction first, and replacement after earthquake". Subjected to the action of frequent earthquakes, the energy dissipating elements cooperate with the rest of the structure in elastic behavior and provide sufficient strength and stiffness; Subjected to the action of moderate earthquakes and rare earthquakes, the energy-dissipating elements enter the plastic stage, providing sufficient energy-dissipating capacity, while the remaining components and accessories still maintain elastic behavior.
(3) For the assembled and beam-column hinged joints with replaceable energy-dissipating elements, when the horizontal length is constant and the ratio of the linear stiffness of the energy dissipation element to the steel beam is approximately 0.7, their seismic performance is in a relatively good state. From the analysis of the 6 groups of tests, it can be concluded that the seismic performance of JD-2 and JD-4 is better than the other groups.
(4) For the assembled and beam-column hinged joints with replaceable energy-dissipating elements, under the condition of constant vertical position and the material properties of energy dissipation elements and steel beams, when the ratio of horizontal length to span is 0.225, their seismic performances are superior. From the analysis of 6 groups of tests, it can be concluded that the seismic performance of JD-4 is better than that of JD-2.
(5) For the assembled and beam-column hinged joints with replaceable energy-dissipating elements, compared with the traditional welded beam-column joints in the steel frame, when the ratio of linear stiffness of energy dissipation element to steel beam is about 0.7 and the ratio of horizontal length to span is about 0.225, they are considered to be a good alternative of it, because the seismic performance is basically close to or beyond that of traditional welded steel frame beam-column joints. From the 6 groups of test analysis, it can be concluded that the seismic performance of JD-4 is better than any other group, and it is closer to the beam-column joint of traditional welded steel frame.
(6) For the assembled and beam-column hinged joints with replaceable energy-dissipating elements, their welding works can be completed in the factory, and there is basically no welding on site, which is conducive to environmental protection; Their beam-column connection nodes are hinged, which is completely controllable in practical applications compared to rigid or semi-rigid connection nodes connected by bolts; After the earthquake damage, the damaged energy-consuming components can be removed to restore normal use, which is convenient and fast, and is conducive to improving the resilience of the city.
Point 2: Lines 88-98: Need to add references next to each author. Remove "Asghari et al. or two authors' last names" Please check the journal's style.
Response 2: (1) ”Scholars such as Asghari proposed……” has been modified to “Scholars such as Abazar [19-20] proposed…… ”; (2) ”Scholars such as H L Hsu proposed……” has been modified to“ Scholars such as H L [21-23] proposed……”
Point 3: Line 105: What is Q345B? Isn't it Chinese Standard? Need to add a reference or standard specification. For example, the ASTM standard is used for materials fabricated or used in the U.S.
Response 3: (1) Q345B is applied in the Chinese standard and it has supplemented the standard specification "GB50017. Standard for design of steel structures. Beijing, China: China construction industry press;2017." (2) ”All components are made of Q345B, high-strength bolts……” has been modified to“ All components are made of Q345B [25], high-strength bolts……”
Point 4: Line 99: Need to recognize some work relevant to recent research for the effort in simulation. There are some efforts to update the model's parameter in in-situ conditions (fields), where repair and replacement are needed.
Jice Zeng and Y.H. Kim, "Stiffness Modification-Based Bayesian Finite Element Model Updating to Solve Coupling Effect of Structural Parameters: Formulations" https://doi.org/10.3390/app112210615
Response 4: It has been supplemented as follows in Line99: At the same time, for a new type of structure with replaceable dissipative elements, when it is repaired after an earthquake, the new BMUA method proposed by Jice Zeng and Young Hoon Kim [24] can be used to update the structural stiffness by considering the coupled effect of mass and stiffness.
Point 5: Line 323: Currently, there is no experimental data regarding joint Type NSF. Is there any estimation from the design equation for the control system's performance?
Response 5: NSF is designed in accordance with the requirements of Chinese code and meets the requirements of seismic design, and its seismic performance evaluation refers to reference 27, which has certain representative significance. The paper adds citations to Chinese code 26 and reference 27, as shown in line 176-180.
Point 6: Figure 6: Caption in Figure 6 shows Bs instead of experimental and analytical results. Need to update.
Response 6: “Bs” has been modified to “Exoeriment and FEA Model”.
Point 7: The authors need to add additional figures to show the elastic yielding point and ultimate strength point with one of the hysteresis curves.
Response 7: The elastic yield point and ultimate yield point have been supplemented in Fig. 6(a) JD-1 hysteresis curve.

Reviewer 2 Report
- A) General remarks
The research presents in this paper a very interesting topic, as well as results that are of wider significance when it comes to the seismic effects on specific civil engineering structures. The paper is concise and clear. The literature in the paper is adequately cited.
However, in the case of literature, it must be pointed out that most of the positions are older than 5 years 25 of 29 and about half are older than 10 years. This is problematic to assess the state of art, especially in the case of the newest achievements in the field.
Additionally, the authors are asked to present more firmly the novelty of the work, especially in the context of current research and other solutions. This may be difficult if not taking care of proper state-of-art presentation and current literature review.
The abstract is well written introducing the basic overview of the paper. However, it is suggested for authors not to use common speech phrases eg. “basically”, “basically consistent” (are the results consistent or not?),” basically close” (is close or not?). It is also suggested not to put technical details like steel type or distances in the abstract. The abstract must stay clear and short and this kind of specificity is not required.
The introduction to the paper is well written and have the most aspects needed to understand the topic. An additional element that is missing is that such a structure as described must be evaluated for a long time to assess its supremacy over other techniques. For the monitoring of the behaviour of structures and buildings and evaluation of dynamic conditions, some probabilistic techniques like PPSD are often used especially in the research centres like LIGO and CERN. For reference, you can use L.Lacny “Application of Probabilistic Power Spectral Density Technique to Monitoring the Long-Term Vibrational Behaviour of CERN Seismic Network Stations”. Additionally, to look into the performance of structures also transfer function approach can be used to see for example see seismic or civil engineering equipment effects on the structure. A similar approach using TF was used here: Guinchard, M., Source based measurements and monitoring of ground motion conditions during civil engineering works for high luminosity upgrade of the LHC, Proceedings of the 26th International Congress on Sound and Vibration, ICSV 2019, (https://www.scopus.com/inward/record.uri?eid=2-s2.0-85084014594&partnerID=40&md5=cd9842078e20cc2d9ba3d36062bc9af4)
When data from both experiment and FEA is compared it would be profitable to add an additional column when % difference is presented (e.g. tab2,3,4,5).
The research design and methods are clearly presented. However, the authors must keep in mind the readers' experience. In some places, the text and explanations are very dense and difficult to follow.
Conclusions are written correctly and in a clear way. However, in the beginning, something is missing or the first sentence is not in correct style (starts with “They were analyzed”).
- B) Item remarks
The common practice is to make a space between the value and the unit (except °C, %), however, authors constantly forget about this rule (from line 133 and farther on). Additionally, the author is asked to keep the convention of units. In some places, they are using what suits them in the other SI units. If possible please unify.
There are many small grammatic errors like in line 177 “steel frames was designed” (should be were). Please check the language before resubmitting. Additionally, the sentence mentioned here is not written in the proper style. There is a dot in the middle and it looks like something is missing or it is just a problem with the style.
Fig.2 would be better if some one-word descriptions of the items seen on the test rig were presented.
Fig.4 please change the axis labels so they are written with the same font as the paper text. At its current state, this plot looks not fully professional for scientific papers. Similarly Fig.6 and 10.
Author Response
Response to Reviewer 2 Comments
1.A) General remarks
Point 1: The research presents in this paper a very interesting topic, as well as results that are of wider significance when it comes to the seismic effects on specific civil engineering structures. The paper is concise and clear. The literature in the paper is adequately cited.
Response 1: Thank you very much for your high evaluation, this review has benefited me a lot.
Point 2: However, in the case of literature, it must be pointed out that most of the positions are older than 5 years 25 of 29 and about half are older than 10 years. This is problematic to assess the state of art, especially in the case of the newest achievements in the field.
Response 2: Thank you very much for the valuable comments! (1) This paper mainly discusses according to the following logics: Firstly, the traditional steel frame has the advantage of being widely used and at the same time the disadvantage of on-site welding of nodes; Secondly, the prefabricated steel structure avoids on-site welding, but cannot be replaced after the earthquake ; Thirdly, three forms of seismic resistance based on recoverable functions are introduced, namely rocking walls, self-recovery function structures and structures with replaceable components; Finally, this paper is based on structures with replaceable components and adopts prefabricated construction. (2) Structures with replaceable components are currently being studied, and each has its own advantages and disadvantages. The node with energy-dissipating elements proposed in this paper is one of the new methods, which has the triple advantages of fully fabricated construction, excellent seismic performance and realizable beam-column hinged in fact; (3) The references cited in this paper are mainly based on the above-mentioned logical and novel research aspects.
Point 3: Additionally, the authors are asked to present more firmly the novelty of the work, especially in the context of current research and other solutions. This may be difficult if not taking care of proper state-of-art presentation and current literature review.
Response 3: Thank you very much for the valuable comments! (1) Structures with replaceable components belong to a large type of earthquake resistance with recoverable functions, and there is currently no such literature review; (2) There are many ways for structures with replaceable components, and it is difficult to carry out its sort. In the future, we will think it over according to your opinion and look for a good direction to classify it and write a relevant literature review.
Point 4: The abstract is well written introducing the basic overview of the paper. However, it is suggested for authors not to use common speech phrases eg. “basically”, “basically consistent” (are the results consistent or not?),” basically close” (is close or not?). It is also suggested not to put technical details like steel type or distances in the abstract. The abstract must stay clear and short and this kind of specificity is not required.
Response 4: (1) The descriptions such as "basic" and "basically consistent " have been modified; (2) The descriptions of steel type and distances have been specified.
Point 5: The introduction to the paper is well written and have the most aspects needed to understand the topic. An additional element that is missing is that such a structure as described must be evaluated for a long time to assess its supremacy over other techniques. For the monitoring of the behaviour of structures and buildings and evaluation of dynamic conditions, some probabilistic techniques like PPSD are often used especially in the research centres like LIGO and CERN. For reference, you can use L.Lacny “Application of Probabilistic Power Spectral Density Technique to Monitoring the Long-Term Vibrational Behaviour of CERN Seismic Network Stations”. Additionally, to look into the performance of structures also transfer function approach can be used to see for example see seismic or civil engineering equipment effects on the structure. A similar approach using TF was used here: Guinchard, M., Source based measurements and monitoring of ground motion conditions during civil engineering works for high luminosity upgrade of the LHC, Proceedings of the 26th International Congress on Sound and Vibration, ICSV 2019, (https://www.scopus.com/inward/record.uri?eid=2-s2.0-85084014594&partnerID=40&md5=cd9842078e20cc2d9ba3d36062bc9af4)
Response 5: Thank you very much for the excellent reviews! (1) In the paper, the performance of fabricated beam-column joints with replaceable energy-dissipating elements is mainly investigated, and the related factors and results of seismic performance are mainly analyzed; (2)Both of PPSD and TF are very good evaluation methods for the long-term effects of the structure, especially for applied projects with new technologies, we will use these methods to evaluate the application of the new technology to actual projects in the next research. (3) In line 132-136, some descriptions of later applications have been added. In the next investigation, when this new type of fabricated beam-column connection with replaceable energy dissipation elements is applied to actual projects, the PPSD method [26] will be used to evaluate the long-term behavior of the structure, using the TF method [27] to examine the effects of seismic or civil engineering equipment on the structure. (4) Two references are added in the paper.
Point 6: When data from both experiment and FEA is compared it would be profitable to add an additional column when % difference is presented (e.g. tab2,3,4,5).
Response 6: In Tables 2 to 5, the percentages for experimental and FEA comparisons have been added, respectively.
Point 7: The research design and methods are clearly presented. However, the authors must keep in mind the readers' experience. In some places, the text and explanations are very dense and difficult to follow.
Response 7: Descriptions of the research design and methods have been simplified, as shown in section “2.1. Test specimens” and “2.2. Test device and loading system”.
Point 8: Conclusions are written correctly and in a clear way. However, in the beginning, something is missing or the first sentence is not in correct style (starts with “They were analyzed”).
Response 8: “They were analyzed on the main seismic performance evaluation factors such as failure mode, strength, stiffness, ductility, and hysteresis curves and energy dissipation capacity through the experimental study and finite element numerical simulation of 6 groups of steel frames with replaceable energy dissipation elements, and compared with the finite element numerical simulation of the traditional welded steel frame. The conclusions are as follows:” has been modified to“ The research was focused on the main seismic performance evaluation factors such as failure mode, strength, stiffness, ductility, and hysteresis curves and energy dissipation capacity through the experimental study and finite element numerical simulation of 6 groups of steel frames with replaceable energy dissipation elements, and compared with the finite element numerical simulation of the traditional welded steel frame. The conclusions are as follows:”
1.B) Item remarks
Point 9: The common practice is to make a space between the value and the unit (except °C, %), however, authors constantly forget about this rule (from line 133 and farther on). Additionally, the author is asked to keep the convention of units. In some places, they are using what suits them in the other SI units. If possible please unify.
Response 9: The amendments are as follows: (1) A space is left between the value and the unit; (2) The SI unit is uniformly adopted.
Point 10: There are many small grammatic errors like in line 177 “steel frames was designed” (should be were). Please check the language before resubmitting. Additionally, the sentence mentioned here is not written in the proper style. There is a dot in the middle and it looks like something is missing or it is just a problem with the style.
Response 10:”At the same time, in order to compare the seismic performance with the traditional all-welded steel frame, a group of steel frames was designed, with the same steel beams and steel columns as the test. in which the nodes are considered as equal-strength welding, and only finite element numerical simulation analysis is performed.” has been modified to“ At the same time, in order to compare the seismic performance of the traditional all-welded steel frame, a group of steel frames is designed. in which the steel beams and columns are the same as those in the test, and the beam-column joints are according to the equal-strength all-welding. In this paper, the steel frame will be only carried out through the finite element numerical simulation analysis.”
Point 11: Fig.2 would be better if some one-word descriptions of the items seen on the test rig were presented.
Response 11: Descriptions of major items such as actuators, dissipative elements and lateral supports have been added to Fig.2.
Point 12: Fig.4 please change the axis labels so they are written with the same font as the paper text. At its current state, this plot looks not fully professional for scientific papers. Similarly Fig.6 and 10.
Response 12: The axis labels for Figure 4 have been changed in the same font as the paper text. Figures 6 and 10 have been modified according to the same principles.

Round 2
Reviewer 1 Report
The authors addressed all the concerns.